# How Hypertension Rates and HIV Treatment Outcomes Compare between Older Females and Males Enrolled in an HIV Treatment Program in Southern Nigeria: A Retrospective Analysis

**DOI:** 10.3390/tropicalmed8090432

**Published:** 2023-08-31

**Authors:** Uduak Akpan, Moses Bateganya, Otoyo Toyo, Esther Nwanja, Chiagozie Nwangeneh, Onwah Ogheneuzuazo, Augustine Idemudia, Ezekiel James, Dolapo Ogundehin, Adeoye Adegboye, Okezie Onyedinachi, Andy Eyo

**Affiliations:** 1Excellence Community Education Welfare Scheme (ECEWS), Uyo 520101, Nigeria; 2Family Health International (FHI 360), Durham, NC 27701, USA; 3US Agency for International Development, Abuja 900211, Nigeria; ejames@usaid.gov (E.J.);

**Keywords:** geriatrics, viral load suppression, retention, sex distribution, HIV/AIDS

## Abstract

Studies show that treatment outcomes may vary among persons living with HIV. To fast-track the attainment of epidemic control across gender and age groups, the Accelerating Control of the HIV Epidemic (ACE-5) Project implemented in Akwa Ibom and Cross Rivers States, Nigeria, examined the hypertension rates and treatment outcomes of older adults living with HIV. The demographic and treatment characteristics of males and females ≥ 50 years living with HIV, who initiated antiretroviral therapy (ART) as of September 2021, were abstracted from medical records across 154 health facilities and community sites in Akwa Ibom and Cross River states, Nigeria. We compared these characteristics by sex using the chi-square test. The log-rank test was used to compare differences in their retention (i.e., being on treatment) and viral suppression (VS) rates [<1000 copies/Ml] in September 2022. Of the 16,420 older adults living with HIV (10.8% of the treatment cohort) at the time of the study, 53.8%, and 99.5% were on a first-line ART regimen. Among the 3585 with baseline CD4 documented (21.8% of the cohort), the median [IQR] CD4 count was 496 [286–699] cells/mm^3^, with more males having lower baseline CD4 than females [13.4% of males vs. 10.2% of females, *p*-value = 0.004]. In total, 59.9% received treatment at out-of-facility locations, with more males receiving treatment in this setting than females [65.7% vs. 54.8% *p*-value < 0.001]. Of those in whom blood pressure was assessed (65.9% of the treatment cohort), 9.6% were hypertensive, with males being less likely to be hypertensive [8.0% vs. 11.1% *p*-value < 0.001] than females. Overall, retention as of September 2022 was 96.4%, while VS was 99.0% and did not differ significantly by sex [retention: *p* = 0.901; VS: *p* = 0.056]. VS was slightly but not significantly higher among females than males (98.8% versus 99.2%; Aor = 0.79, 95%CI = 0.58–1.10, *p* = 0.17). Although older males and females living with HIV had similar treatment outcomes, hypertension screening was suboptimal and could impact long-term morbidity and mortality. Our study emphasizes the need to integrate noncommunicable disease screening and the management of hypertension in the care of older persons living with HIV.

## 1. Introduction

The global scale-up of antiretroviral therapy (ART) has led to improved survival of people living with the human immunodeficiency virus (HIV), leading to a growing number of older adults living with HIV [1,2]. The decline in HIV-related deaths, by 52% between 2010 and 2021 [3], due to the availability of more efficacious, less toxic treatment options means that people with HIV (PWH) now live longer and have a better quality of life [4]. Globally, people living with HIV (PLHIV) aged 50 years and above account for 13% of adults living with HIV [5]. Sub-Saharan Africa has 60% of older PLHIV [2], and, in Nigeria, 13% of PLHIV are 50 years and older [5,6]. The number of older adults living with HIV is projected to increase because of innovative treatment strategies that have resulted in superior treatment outcomes and a higher life expectancy for PLHIV [7,8].

However, aging increases the risk of non-infectious morbidity, and comorbidities have been associated with chronic inflammation from HIV infection and adverse effects from long-term ART [9,10,11,12]. Individuals living with HIV who have other health conditions or comorbidities tend to have poorer treatment outcomes [11,13]. Furthermore, treatment outcomes can also vary based on gender. Several studies have documented gender differences in treatment outcome and mortality among PLHIV [14,15,16].

In Nigeria, ART coverage has significantly improved from 63% in 2019 to 89% in 2020, and over 25% of all adults living with HIV are categorized as older adults (i.e., age ≥ 50 years of age). While the sociodemographic features and clinical outcomes of PLHIV have been well described [17,18,19], there is limited knowledge regarding the gender-specific differences in the treatment outcome of older adults living with HIV, certainly in Nigeria and other resource-limited settings. The Accelerating Control of the HIV Epidemic (ACE-5) Project funded by the US Presidents Plan for AIDS Relief (PEPFAR) through the United States Agency for International Development (USAID), implemented by Excellence Community Education Welfare Scheme (ECEWS) in Akwa Ibom and Cross Rivers States, Nigeria, aims to fast track the attainment of epidemic control across gender and different age groups. To improve treatment outcomes across all populations, the ACE-5 project utilizes differentiated service delivery models at health facilities or community sites.

We conducted an analysis of program data to determine the burden of hypertension in males and females living with HIV in southern Nigeria who were older than 50 years, and we described the clinical characteristics, retention in care, and viral suppression. Our hypothesis was that there are differences between genders in the proportion with hypertension and that HIV treatment outcomes also differ among older female and male adults receiving ART.

## 2. Methods

### 2.1. Setting

The ACE-5 project was implemented in 34 local government areas (LGAs) in Akwa Ibom and Cross River States, Nigeria. ART services were provided through 154 health facilities, 95 community pharmacies, and 68 community ART management (CAM) teams comprising community mobilizers (for community entry and mapping), counsellor testers for testing, clinicians (for clinical management), and case managers (for case management and follow-up). To provide person-centered care to older PLHIV, alongside the standard care provided within the health facilities, the ACE 5 project also maintained the three community-based models (community pharmacy ART refill program (CPARP), community ART refill groups (CARGs), community ART refill clubs (CARCs)) and other decentralized drug distribution models, as previously described by Sanwo O. et al. [20].

### 2.2. Study Design/Population

This retrospective cohort study analyzed program data for PLHIV (≥50 years) who initiated ART prior to September 2021 in 154 health facilities in Akwa Ibom and Cross River States in southern Nigeria.

### 2.3. Data Collection

For this study, de-identified data were extracted from Lafiya Management Information System (LAMIS), an electronic medical record database—described previously—housing routine programmatic data collected from PLHIV who access services at supported health facilities [21]. These service delivery data are collected using standardized paper-based forms at each patient encounter and then entered into LAMIS by facility staff. At the end of each day, the data are summarized and validated internally. Summary reports submitted to the project are compared with source documents such as registers and other patient-monitoring forms quarterly to ensure consistency. Routinely during the data quality assurance process, if discrepancies are observed in the data, then reasons for the discrepancies are ascertained and noted. The data in LAMIS are then adjusted to ensure consistency with the source document.

For this analysis, data were abstracted for each patient record at baseline, including sociodemographic (sex, age, marital status, education) characteristics, residence (classified rural or urban), ART regimen (1st-line ARVs: Tenofovir + Lamivudine + Dolutegravir or Abacavir + Lamivudine + Dolutegravir; 2nd-line ARVs: Tenofovir + Lamivudine + Atazanavir/ritonavir, Abacavir + Lamivudine + Atazanavir/ritonavir, Zidovudine + Lamivudine + Atazanavir/ritonavir, or Abacavir + Lamivudine + Atazanavir/ritonavir), the presence of hypertension (documented systolic blood pressure (BP) >  140 mmHg or diastolic BP > 90 mmHg), and the use of antihypertensive medication. Also extracted were ART status at the end line in September 2022 and recent viral load results. The extracted data contained no patient names or other personal information that could be used to identify individual patients, and they were subjected to internal consistency checks and assessed for outliers and wrong entries, which were removed prior to analysis.

Time-based cohorts of eligible individuals were created based on the year of ART initiation, which also aligned with the PEPFAR program year that starts on October 1 and runs until September 30th of the subsequent year. This yielded seven cohorts: before October 2015, October 2015–September 2016, October 2016–September 2017, October 2017–September 2018, October 2018–September 2019, October 2019–September 2020, and October 2020–September 2021. We excluded individuals transferred in during the study follow-up period.

### 2.4. Outcomes of Interest

Documentation of high blood pressure and retention for the 12-, 24-, 36-, 48-, 60-, 72-, and >72-months-on-ART cohorts was determined in September 2022. PLHIV having no clinic visit within >28 days from the last expected drug refill were considered lost-to-follow-up. Other outcomes were “stopped” (if the individual patients stopped treatment), “dead” (if a death was reported by relatives or certified by a clinician), or “transferred-out” (if patient was transferred to another health facility within or outside the community); otherwise, the patients were considered “active”.

Viral suppression status was determined based on the client’s last viral load in September 2022 and categorized as “suppressed (HIV RNA threshold less than 1000 copies/mL)” or “unsuppressed (HIV RNA threshold greater or equal 1000 copies/mL)” following national and WHO guidelines [22,23]. Viral suppression was further stratified based on three (3) limits of detection as follows: 0–40 copies/mL; 41–200 copies/mL; and 201–999 copies/mL. The Roche COBAS^®^ AmpliPrep, Abbott, and Hologic Panther ^®^ are the platforms used for VL analysis and have a low detection limit of <40 copies/mL, which informed the various classifications.

### 2.5. Data Analysis

Descriptive statistics (frequencies) were used to summarize categorical variables. Kaplan–Meier was used to assess retention rate (i.e., being active), with an apriori expectation of a 95% PEPFAR retention threshold [24]. The log-rank test was used to assess differences in retention rates by sex and other client characteristics. Differences in proportions of individuals who had hypertension and those who were virally suppressed were compared using the chi-square test and Fisher’s exact test where applicable. All analyses were conducted using SPSS version 24 and considered significant with a *p*-value of <0.05.

### 2.6. Ethical Considerations

Permission to analyze these program data was obtained from the Health Research Ethics Committee (HREC) in Akwa Ibom state. Patient informed consent was not required because only routine, anonymous, operational monitoring data were collected and analyzed.

## 3. Results

A total of 16,420 older adults living with HIV (10.8% of the treatment cohort) initiated ART as of September 2021. The majority, 53.8%, were females. The percentages of older males who started on ART in each cohort year increased by 14.0 percentage points from 39.1 percentage points in 2015 to 53.1 percentage points in 2021 (Table 1).

Most of the enrolled cohort (91.4%) were 50–65 years, 59.9% were identified in the community, and 60.6% resided in rural areas. Above half (54.1%) had secondary education, and the majority (69.7%) were married. A greater proportion of the older adults living with HIV (99.5%) were on first-line ART regimens. Blood pressure (BP), although a vital sign expected to be recorded at each encounter, was not consistently recorded, with only 10,828 (65.9%) of all files having a measurement. Of those with BP documentation, 9.6% were reported to be hypertensive. Of 1041 with hypertension, 627 (60.2%) were women. A significantly higher proportion of females (11.1%) than males (8.0%) had hypertension (*p* < 0.001). Other comorbid conditions were not consistently documented and are not reported. CD4 testing was also either not consistently conducted or reported, with only 3585 or 21.8% of the cohort having files with a CD4 result. Of those with documentation, the median [interquartile range] CD4 count was reported to be 496 cells/mm^3^ [286–699], and, of those with a CD4, 11.4% [13.4 among males and 10.2% among females] had CD4 less than 200 cells/mm^3^. When compared by sex, there was a significantly higher number of males identified and commenced on ART in the community than females (65.7% versus 54.8%, *p* < 0.001). More males had secondary education (61.5%) when compared to females (47.6%) (Table 2).

Overall, the viral suppression rate was 99.0%, slightly but not significantly higher among females than males (98.8% versus 99.2%; aOR = 0.79, 95%CI = 0.58–1.10, *p* = 0.17) (Table 3). In addition, viral suppression was stratified by various thresholds. The majority (93.3%) of the reviewed cohort achieved undetectable VL between <40 copies/mL; 3.9% had VL ranges of 41–200 copies/mL, while 1.9% had VL ranges of 201–999 copies/mL. There were no significant differences when these VL thresholds were compared by sex [0–40 copies/mL: aOR = 0.93, 95%CI = 0.81–1.06, *p* = 0.26; 41–200 copies/mL: aOR = 1.03, 95%CI = 0.88–1.21, *p* = 0.70; 201–999 copies/mL: aOR = 1.16, 95%CI = 0.93–1.46, *p* = 0.19].

Viral suppression was slightly lower among those who received ART in the facility (98.4%) compared to those who received ART in the community (99.5%), lower in urban dwellers (98.5%) than in rural dwellers (99.4%), and lower among those separated from a spouse (98.7%) than those married (99.1%) or single (99.2%). By comparing regimen type, viral suppression was lower among those on a fixed-dose combination of Tenofovir + Lamivudine + Dolutegravir (99.1%) than among those on the Abacavir + Lamivudine + Dolutegravir (98.4%), and among those with hypertension (99.4%) than among those without hypertension (99.2%). All differences were not significant.

Females who received ART in the facility had better suppression than males (females, 98.8%: males, 97.7%; *p* = 0.01); similarly, females on first-line ARVs had higher suppression than males (females, 99.2%: males, 98.9%; *p* = 0.02). In the subgroup with low CD4 (<200 cells/mm^3^), females achieved higher viral suppression than males (females, 99.5: males, 94.1%; *p* ≤ 0.001). However, there was no sex difference in viral suppression among those with hypertension (Table 4).

Overall, the retention rate was 96.4% (Table 5). Retention rates were consistently high, except for in the 36-month cohort in which it was below the 95% PEPFAR threshold. There was no significant difference in retention by sex (females, 96.4% versus males, 96.4%, *p* = 0.90).

Females who received ART in the health facility were more likely than males to be retained on treatment [female: 95.4% vs. male: 93.9%; *p*-value = 0.008]. Similarly, females residing in rural areas were more likely than males to be retained on ART [females: 97.9% vs. males: 98.5%; *p*-value = 0.02]. There was no difference in retention between males and females based on age category, educational level, regimen line and type, hypertension status, or baseline CD4 status (Table 6).

## 4. Discussion

Our study determined the burden of hypertension and compared characteristics and HIV treatment outcomes among males and females older than 50 years living with HIV in southern Nigeria. In this cohort, 96.4% were retained on antiretroviral therapy, and up to 99.0% achieved viral load suppression. These important program and clinical outcomes were similar for both males and females. However, we found poor rates of screening or documentation of hypertension, with only 65.9% having been assessed for this important comorbidity.

Overall, we found more older females on treatment than men, consistent with the Nigerian epidemic [6,25]. However, the proportion of older males to females who started on ART increased by 14.0 percentage points from 39.1 percentage points in 2015 to 53.1 percentage points in 2021. Also, a greater proportion of older males living with HIV were identified in the community and continued to receive community-based care. We present these findings, and although obtaining them was not the objective of this study, the setting could have a bearing on access to or future planning of services such as hypertension screening. Several studies have highlighted the challenges of finding males and retaining them on ART [26,27], thus, community testing, ART initiation and maintenance in the community could be a viable strategy for ending AIDS. We attribute the higher male case finding to a community-based ART surge intervention that was implemented from 2019 to 2021 [28]. During the surge, HIV testing was provided by outreach teams at community hot spots during convenient times, the testing of sexual partners of newly diagnosed HIV individuals, and same-day ART initiation in the community were implemented at scale, resulting in high case finding, treatment initiation, and better treatment outcomes [29]. Indeed, community-based ART programs have been shown to make treatment more accessible and affordable [29,30], which may account for the comparable treatment outcomes and the absence of sex differences in retention and viral suppression seen in this study. This finding, as with that of other studies [30,31], highlights the role of the community in the delivery of care for men. Thus, policymakers and program implementers should consider these findings when designing interventions to cater to the needs of this population.

In this study, the prevalence of hypertension among older adults living with HIV was 9.6%, lower than the 21% reported in a different Nigerian geriatric HIV cohort [13], the 32% reported in sub-Saharan Africa, and the 42% reported globally for a similar age group (≥50 years) [32]. More females were hypertensive than males, which is in keeping with similar studies that showed a two-to-five-fold increase in the risk of hypertension for females [33,34]. What is concerning, however, is that only 61.7% of the older adults in this cohort had access to regular blood pressure monitoring, a potentially missed opportunity for identifying those at risk of morbidity and mortality from hypertension. This finding strengthens the evidence of the sub-optimal health system response to non-communicable disease prevention and management. With the increasing mortality from non-communicable diseases, a differentiated approach that integrates the principles of geriatric medicine in HIV programs is needed to manage older persons living with HIV [35]. Several studies described integration models suitable for low-income countries to provide these services, focusing on early detection through routine screening and subsequent care, even at the level of primary health care [36,37]. Nonetheless, concerns around human resources, infrastructure and equipment, and the supply chain still abound.

Retention and viral load suppression were high in this cohort of older PLHIV and were similar among males and females, unlike in other studies that have highlighted challenges in keeping males engaged in HIV treatment programs [38]. Our findings are in line with those of regional studies. A cohort study conducted in three countries in Central Africa showed that older adults were 1.6 times more likely to be adherent to ART than younger age groups [39]. Another study in Kenya observed a better retention rate among older adults than among younger adults [40]. These studies reporting higher rates of retention and virologic suppression among older patients have attributed this association to better adherence to medication by the older age groups due to their experience in taking medication for other chronic non-HIV conditions [41,42]. Our study did not show superior retention in the group with hypertension, but the number with blood pressure monitoring was very small.

Our finding of higher rates of viral suppression in females than in males in a small subpopulation of individuals with low baseline CD4 was surprising and warrants further inquiry. The low viral suppression in men in in the proportion with more advanced HIV is likely due to other concurrent comorbidities—beyond the data that we analyzed—that are probably more prevalent among men.

Our study, like all retrospective studies, had some limitations. Firstly, variations exist among older adults living with HIV beyond the sociodemographic characteristics that we had access to for use in this analysis. For example, a review of retrospective data did not allow us to determine the reasons for the lower prevalence of hypertension in this setting, thus needing further research. Secondly, we used routine program data that have inherent weaknesses, including missingness and inconsistency in their recording.

However, this study has some notable strengths and contributions to service delivery design. We used routine program data from a large PEPFAR-funded program that provided care to a wide variety of patients from different localities. While not generalizable, some lessons can be applied to similar resource settings like ours.

## 5. Conclusions

In this large cohort of older adults living with HIV receiving ART, assessment and/or documentation of hypertension screening was poor, but hypertension rates were lower than in similar settings. However, retention in the HIV program and viral suppression rates were above the PEPFAR retention benchmark and UNAIDS target of 95% of PLHIV on ART having viral suppression. There were no differences in viral suppression between males and females, except for among those with advanced HIV, highlighting the successful engagement of older PLHIV in our HIV program. However, the low access to hypertension screening and access to CD4 testing at ART initiation are of concern. To address this, policymakers should explore an integrated approach that merges elements of geriatric medicine principles with HIV programs. This holistic strategy would ensure comprehensive care for older individuals living with HIV, with a particular emphasis on routine screening, early detection, and ongoing care, even at the primary healthcare level. The elevated prevalence of hypertension in older females warrants further investigation into causes and control measures. Our study findings can inform the design and implementation of a package of care for older persons living with HIV.

## Figures and Tables

**Table 1 tropicalmed-08-00432-t001:** Number of older adults living with HIV started on ART before September 2021, disaggregated by year of initiation * and sex, in southern Nigeria (n = 16,420).

Period	Sex				
	Male	Frequency	Female	Frequency	Total
<October 2015	767	39.1%	1194	60.9%	1961
October 2015–September 2016	292	37.8%	481	62.2%	773
October 2016–September 2017	268	35.1%	496	64.9%	764
October 2017–September 2018	326	39.6%	498	60.4%	824
October 2018–September 2019	787	40.8%	1140	59.2%	1927
October 2019–September 2020	2458	48.1%	2653	51.9%	5111
October 2020–September 2021	2688	53.1%	2372	46.9%	5060
Total	7586	46.2%	8834	53.8%	16,420

* Cohorts were categorized by program year October–September.

**Table 2 tropicalmed-08-00432-t002:** Demographic characteristics of older adults living with HIV who initiated ART before September 2021 in southern Nigeria, disaggregated by sex (n = 16,420).

		Total	Male	Female	*p*-Value (Chi-Square)
Age group					
	50–65 years	15,007 (91.4%)	6936 (91.4%)	8071 (91.4%)	0.88
	≥65 years	1413 (8.6%)	650 (8.6%)	763 (8.6%)	
	Missing = 0				
Care delivery point					
	Community	9831 (59.9%)	4986 (65.7%)	4845 (54.8%)	<0.001
	Facility	6589 (40.1%)	2600 (34.3%)	3989 (45.2%)	
	Missing = 0				
LGA of residence					
	Urban	6462 (39.4%)	2931 (38.6%)	3531 (40.4%)	0.08
	Rural	9958 (60.6%)	4655 (61.4%)	5303 (60.0%)	
	Missing = 0				
Education					
	None	2083 (13.9%)	774(11.0%)	1309 (16.3%)	<0.001
	Primary	4813 (32.0%)	1928 (27.5%)	2885 (36.0%)	
	Secondary	8130 (54.1%)	4317 (61.5%)	3813 (47.6%)	
	Missing = 1394				
Marital status					
	Single	1179 (7.4%)	608 (8.2%)	571 (6.7%)	<0.001
	Married	11,070 (69.7%)	5734 (77.7%)	5336 (62.9%)	
	Separated	3624 (22.8%)	1041 (14.1%)	2583 (30.4%)	
	Missing = 547				
Regimen line					
	1st-Line ARVs	16,338 (99.5%)	7543 (99.4%)	8795 (99.6%)	0.256
	2nd-Line ARVs	82 (0.5%)	43 (0.6%)	39 (0.4%)	
	Missing = 0				
Current ART (1st line only)					
	Tenofovir + Lamivudine + Dolutegravir	16,338 (99.5%)	7543 (99.4%)	8795 (99.6%)	0.256
	Abacavir + Lamivudine + Dolutegravir	82 (0.5%)	43 (0.6%)	39 (0.4%)	
Diagnosed with hypertension					
	No	9787 (90.4%)	4751 (92.0%)	5036 (88.9%)	<0.001
	Yes	1041 (9.6%)	414 (8.0%)	627 (11.1%)	
	Missing BP = 5592				
CD4 at initiation					
	≥200 cells/mm^3^	3176 (88.6%)	1242 (86.7%)	1934 (89.8%)	0.004
	<200 cells/mm^3^	409 (11.4%)	190 (13.4%)	219 (10.2%)	
	Median	496 cells/mm^3^ [286–699]	431 cells/mm^3^ [266–627]	504 cells/mm^3^ [304–737]	
	Missing = 12,835				

**Table 3 tropicalmed-08-00432-t003:** Viral Suppression Rates for Older Adults Living with HIV (n = 16,286) in Southern Nigeria as of September 2022, Disaggregated by Months on ART.

	Years on ART	>72 Months	72 Months	60 Months	48 Months	36 Months	24 Months	12 Months	Total
Male	Number who had VL test	754	292	268	324	784	2453	2656	7531
	Number suppressed	742	283	262	321	761	2439	2636	7444
	% suppressed	98.4%	96.9%	97.8%	99.1%	97.1%	99.4%	99.2%	98.8%
	*0*–*40 copies/mL*	676 (89.7%)	255 (87.3%)	235 (87.7%)	287 (88.6%)	728 (92.9%)	2317 (94.5%)	2527 (95.1%)	7025 (93.3%)
	*41*–*200 copies/mL*	46 (6.1%)	18 (6.2%)	19 (7.1%)	23 (7.1%)	22 (2.8%)	77 (3.1%)	80 (3.0%)	285 (3.8%)
	*201*–*999 copies/mL*	20 (2.7%)	10 (3.4%)	8 (3.0%)	11 (3.4%)	11 (1.4%)	45 (1.8%)	29 (1.1%)	134 (1.78%)
Female	Number who had VL test	1159	480	496	497	1138	2646	2339	8755
	Number suppressed	1149	474	491	490	1125	2630	2326	8685
	% suppressed	99.1%	98.8%	99.0%	98.6%	98.9%	99.4%	99.4%	99.2%
	*0*–*40 copies/mL*	1070 (92.3%)	439 (91.5%)	451 (90.9%)	453 (91.2%)	1058 (93.0%)	2492 (94.2%)	2204 (94.2%)	8167 (93.3%)
	*41*–*200 copies/mL*	60 (5.2%)	21 (4.4%)	26 (5.2%)	25 (5.0%)	43 (3.8%)	92 (3.5%)	80 (3.4%)	347 (4.0%)
	*201*–*999 copies/mL*	19 (1.6%)	14 (2.9%)	14 (14 (2.8%)	12 (2.4%)	24 (2.1%)	46 (1.7%)	42 (1.8%)	171 (2.0%)
Total	Number who had VL test	1913	772	764	821	1922	5099	4995	16,286
	Number suppressed	1891	757	753	811	1886	5069	4962	16129
	% suppressed	98.8%	98.1%	98.6%	98.8%	98.1%	99.4%	99.3%	99.0%
	*0*–*40 copies/mL*	1746 (91.3%)	694 (89.9%)	686 (89.8%)	740 (90.1%)	1786 (92.9%)	4,809 (94.3%)	4731 (94.7%)	15236 (93.3%)
	*41*–*200 copies/mL*	106 (5.5%)	39 (5.1%)	45 (5.9%)	48 (5.9%)	65 (3.4%)	169 (3.3%)	160 (3.2%)	632 (3.9%)
	*201*–*999 copies/mL*	39 (2.0%)	24 (3.11%)	23 (2.8%)	23 (2.8%)	35 (1.8%)	91 (1.8%)	71 (1.4%)	307 (1.9%)

**Table 4 tropicalmed-08-00432-t004:** Viral suppression rates for older adults living with HIV (n = 16,343) in southern Nigeria as of September 2022, disaggregated by client characteristics.

		Viral Load Suppression; n = 16,129			
		Total	Male	Female	*p*-Value (Chi-Square)
Age group					
	50–65	14750 (99.0%)	6811 (98.9%)	7939 (99.2%)	0.06
	≥65	1379 (99.1%)	633 (98.6%)	746 (99.5%)	0.09
Care delivery point					
	Community	9733 (99.5%)	4937 (99.5%)	4796 (99.5%)	0.64
	Facility	6396 (98.4%)	2507 (97.7%)	3889 (98.8%)	**0.01**
LGA of residence					
	Urban	4823 (98.5%)	2243 (98.4%)	2580 (98.5%)	0.12
	Rural	9415 (99.4%)	4459 (99.2%)	4956 (99.6%)	0.06
Education					
	None	1781 (99.5%)	723 (99.4%)	1058 (99.5%)	0.95
	Primary	4724 (98.9%)	2829 (98.9%)	1895 (98.9%)	0.94
	Secondary	7991 (99.0%)	4230 (98.6%)	3761(99.3%)	**0.002**
Marital status					
	Single	1164 (99.2%)	601 (99.3%)	563 (99.1%)	0.67
	Married	10,897 (99.1%)	5624 (98.8%)	5273 (99.5%)	**<0.001**
	Separated	3533 (98.7%)	1021 (98.7%)	2512 (98.7%)	0.99
Regimen line					
	1st-Line ARVs	16,051 (99.1%)	7403 (98.9%)	8648 (99.2%)	**0.02**
	2nd-Line ARVs	78 (95.7%)	41 (95.3%)	37 (94.9%)	0.92
Current ART (1st line only)					
	Tenofovir + Lamivudine + Dolutegravir	15,928 (99.1%)	7357(98.9%)	8571 (99.2%)	**0.03**
	Abacavir + Lamivudine + Dolutegravir	123 (98.4%)	46 (95.8%)	77 (100%)	0.07
Hypertension					
	No	9677 (99.4%)	4698 (99.3%)	4979 (99.5%)	0.26
	Yes	1018 (99.2%)	407 (99.3%)	611 (99.2%)	0.89
CD4 at initiation					
	≥200 cells/mm^3^	3097 (98.7%)	1213 (98.5%)	1884 (98.9%)	0.33
	<200 cells/mm^3^	390 (97.0%)	176 (94.1%)	214 (99.5%)	**0.02**

**Table 5 tropicalmed-08-00432-t005:** Retention rates among older adults living with HIV on ART in southern Nigeria as of September 2022, disaggregated by client cohort.

ART Initiation Cohort	Retention (N = 16,259)					
	Male (n/N = 7254/7527)			Female (n/N = 8420/8732)		
	Total N	Attrition	Retention Rate (%)	Total N	Attrition	Retention Rate (%)
>72 months	761	20	97.4%	1186	28	97.6%
72 months	288	13	95.5%	470	16	96.6%
60 months	265	18	93.2%	492	16	96.7%
48 months	321	25	92.2%	487	21	95.7%
36 months	776	56	92.8%	1122	62	94.5%
24 months	2436	60	97.5%	2628	76	97.1%
12 months	2680	81	97.0%	2347	93	96.0%
Overall	7527	273	96.4%	8732	312	96.4%
Log-rank (Mantel–Cox) test = 0.015, *p*-value = 0.901						

Excludes ART transfer-out.

**Table 6 tropicalmed-08-00432-t006:** Retention rates among older adults living with HIV on ART in southern Nigeria as of September 2022, disaggregated by characteristics of older adults.

		Individuals Retained (%)			
		Total	Male	Female	*p*-Value (Chi-Square)
Age group					
	50–65	14,375	6655 (96.7%)	7720 (96.7%)	0.81
	≥65	1299	599 (93.3%)	700 (93.2%)	0.95
Care delivery point					
	Community	9534	4851 (97.6%)	4683 (97.2%)	0.23
	Facility	6140	2403 (93.9%)	3737 (95.4%)	**0.008**
LGA of residence					
	Urban	5928	2677 (92.9%)	3251 (94.2%)	0.03
	Rural	9746	4577 (98.5%)	5169 (97.9%)	**0.02**
Education					
	None	1992	744 (96.6%)	1248 (96.5%)	0.90
	Primary	4610	1854 (96.8%)	2756 (96.7%)	0.86
	Secondary	7735	4110 (96.1%)	3625 (96.2%)	0.81
Marital status					
	Single	1130	581 (96.5%)	549 (97.2%)	0.52
	Married	10,580	5493 (96.5%)	5087 (96.5%)	0.97
	Separated	3432	982 (95.3%)	2450 (96.0%)	0.37
Regimen line					
	1st-Line ARVs	15,600	7217 (96.4%)	8383 (96.4%)	0.90
	2nd-Line ARVs	74	37 (86.0%)	37 (94.9%)	0.22
	3rd-Line ARVs				
Current ART (1st line only)	Tenofovir + Lamivudine + Dolutegravir	15,495	7178 (96.5%)	8317 (96.5%)	0.90
	Abacavir + Lamivudine + Dolutegravir	105	39 (79.6%)	66 (88.0%)	0.20
Hypertension					
	No	9514	4620 (97.8%)	4894 (98.0%)	0.58
	Yes	1081	406 (98.8%)	612 (98.6%)	0.75
CD4 at initiation					
	≥200 cells/mm^3^	3024	1181 (96.1%)	1843 (96.6%)	0.49
	<200 cells/mm^3^	366	166 (88.3%)	200 (93.0%)	0.12

## Data Availability

The data that support the findings of this study are available on request from the corresponding author.

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
