# Peer review of "How Hypertension Rates and HIV Treatment Outcomes Compare between Older Females and Males Enrolled in an HIV Treatment Program in Southern Nigeria: A Retrospective Analysis"

_tropicalmed, 2023, doi:10.3390/tropicalmed8090432_

Round 1

Reviewer 1 Report

Thanks for the opportunity to review this well written manuscript with important data. This retrospective cohort study analyzed program data for PLHIV (≥50 years) that initiated ART as of September 2021 in 154 health facilities in Akwa Ibom and Cross River States in Southern Nigeria. There is limited knowledge regarding the gender-specific differences in the treatment outcome of older adults living with HIV. Accelerating Control of the HIV Epidemic (ACE-5) Project funded by PEPFAR through USAID, implemented by ECEWS in Akwa Ibom and Cross Rivers States, Nigeria aimed to fast track the attainment of epidemic control across gender and different age groups.

They looked at the clinical characteristics, retention in care, and viral suppression of older men and women. The authors hypothesized that there were differences between genders in these outcomes among older adults who are receiving antiretroviral therapy. In this cohort, 96.4% were retained on antiretroviral therapy, and up to 98.7% achieved viral load suppression. The outcomes were similar for both males and females. 

The authors described the limitations of the study, acknowledging that variations exist among older adults living with HIV beyond the socio-demographic characteristics and that the routine program data that they used had missingness and inconsistency in its recording. However, the study has some notable strengths and contributions to service delivery, as it can help design and implement a package of care for older persons living with HIV that can be applied to similar resource settings like ours.

I have one minor comment for the authors and think it would be interesting to know more detailed data on viral load suppression. As optimal viral suppression is defined as a confirmed HIV RNA level below the lower limit of detection (LLD) of available assays (generally <20 copies/mL, depending on the assay used), would it be possible to stratify data on the following: < LLD, >LLD and < 200 copies/mL, and > 200 copies/mL and < 1000 copies/mL? Additionally, I suggest adding the threshold for HIV RNA to the methodology section of the manuscript.

Reviewer 2 Report

Reviewer’s comments

“How do outcomes compare between older women and men en-2 rolled in an HIV treatment program in Southern Nigeria: A retrospective analysis” by Akpan U et al.,

The study is interesting and informative.

I have several comments as follows.

Title

-For “older women and men” in the title, I think the authors mean older biological females and males, which might be more appropriate academic terms.

Abstract

-It might be better if the authors could add 1-2 sentences of background or context of the study, before the first method sentence.

-Line 24: the unit of CD4 here and everywhere throughout the manuscript should be “cells/mm3”, not cells/mL.

-Line 30-31: the conclusion was irrelevant to the study findings. Please consider revising i.e. recommendations on how HIV care be different for males and females, or no need to differentiate.

Methods

-Line 80-81: Inclusion criteria were PLHIV >/= 50 years who initiated ART in 154 health facilities. However, the results reported more men received ART in communities and had lower rate of virologic suppression. Please provide more details on ART care system, were they sent out to communities after treatment initiation? Who was eligible for sending out? and how they were followed? Were the same guideline/protocol used (frequency of CD4 and VL testing, vital signs measurement etc.) ?

Results

-The first paragraph on page 3, lines 133-138 presented the results which did not within the study objectives (proportion of men and all PLHIV started ART during 2015-2021). Where this data came from? Please add relevant study objectives and the analysis.

-Page 3, Line 137-138: Th author mentioned “proportion of older males to females”. While they were percentage of males who started ART in each year. Please revise.

-Page 4, Line 157-158: More males had secondary education could mislead as it is not clear if females had higher or lower education.

-Page 6, line 176: what did HTN mean?

Table 2

-Row 4: What is LGA mean?

-Row 7: Could the authors provide details on the 1st, 2nd, and 3rd regimen? What were they? The number was zero for the 3rd regimen, so it could be removed from the table.

-Row 10: CD4 at initiation in columns 2-5, the unit should be cells/mm3, not cells/mL or copies/mL.

Table 4

-Please correct the unit of CD4.

Table 6

-Row 2: Age group should be 50-<65 and >/= 65 years.

-Please correct the unit of CD4.

Discussion

-As more men were followed out of health facilities, they were more likely to have blood pressure measurements and diagnosis of hypertension. Was this among the reasons for the lower prevalence of hypertension?

-Page 9, line 223: The author compared the prevalence of hypertension with the Nigerian Geriatric cohort and a few studies in different regions. Were their population aged older?

-The conclusion can be improved to reflect significant study findings. What are the study implications for this country and for other similar settings? 
